# Computational model of alpha 7 nicotinic acetylcholine receptor in thalamic reticular nucleus neurons and their involvement in network states

Aman Ullah[1,2]*, Karen Therrien[1], John R. Cressman[3], Nadine Kabbani[1,2]

**1** Interdiscplinary Program in Neuroscience, George Mason University, Fairfax, Virginia, United States of America, **2** School of Systems Biology, George Mason University, Fairfax, Virginia, United States of America, **3** Department of Physics and Astronomy, George Mason University, Fairfax, Virginia, United States of America

* aullah3@gmu.edu

## Abstract

Human 15q13.3 microdeletion syndrome (15q13mds) is a genetic disorder caused by a heterozygous deletion of multiple genes, including the CHRNA7 gene, which encodes the α7 nicotinic acetylcholine receptor (α7 nAChR). This condition is associated with significant neurodevelopmental impairments and an increased risk of seizures, with studies indicating reduced α7 nAChR expression in affected individuals. To explore the role of α7 nAChR activity, we developed computational models of the thalamic reticular nucleus (TRN), a brain region critical for regulating thalamocortical (TC) oscillations involved in epilepsy and sleep-wake states. Using a single-compartment kinetic model of a TRN neuron embedded in a simplified thalamic network model, we demonstrate that α7 nAChR activity is necessary to modulate neuronal firing, through calcium regulation, and produce distinct wake and sleep-like states with the network. These findings suggest that α7 nAChR activity in the TRN modulates TC oscillations between sleep and wake states and can contribute to absence seizures in 15q13mds and other neurodevelopmental disorders.

## Introduction

Epilepsy encompasses a group of acquired and genetic syndromes characterized by the chronic occurrence of seizures. Human 15q13.3 microdeletion syndrome (15q13mds) is one such developmental genetic disorder, resulting from the heterozygous deletion of at least six genes on chromosome 15q13.3, with an estimated prevalence of ~0.03% in the general population [1,2]. 15q13mds is associated with a spectrum of neuropsychiatric manifestations, including intellectual disabilities, behavioral abnormalities, and neurodevelopmental disorders such as autism spectrum disorder (ASD), schizophrenia [3,4], and idiopathic generalized epilepsy (IGE) [5].

**Data availability statement:** All relevant data are within the paper.

**Funding:** The author(s) received no specific funding for this work.

**Competing interests:** The authors have declared that no competing interests exist.

Among the deleted genes, CHRNA7, encoding the α7 nicotinic acetylcholine receptor (α7 nAChR) subunit, has emerged as a critical focus in research on 15q13mds and IGE [6,7]. The α7 nAChR is a homopentameric channel with high calcium conductance [8]. The loss of an CHRNA7 allele has been shown to significantly reduce α7 subunit protein expression in brain [9,10]. Postmortem studies of the brain of individuals with 15q13mds confirms a 50% reduction in α7 mRNA [11], while CHRNA7 knockout mice exhibit an increased risk of epilepsy [12,13].

Thalamocortical (TC) circuits, which play a central role in wakefulness, attention, and sleep, are strongly implicated in the pathophysiology of epilepsy. These circuits are modulated by acetylcholine (ACh) originating from cholinergic nuclei in the brainstem and basal forebrain [14,15]. Within the thalamus, the thalamic reticular nucleus (TRN), a cluster of GABAergic neurons, provides inhibitory input to TC relay neurons [16]. Direct cholinergic modulation of TRN neurons occurs through both muscarinic ACh receptors (mAChRs) and nAChRs, leading to shifts between tonic and burst firing modes [16]. ACh also influences acceleration-deceleration bursting within TC oscillatory states [17–19].

T-type voltage-gated calcium channels in TRN neurons are critical for maintaining low-threshold spike (LTS) responses at hyperpolarized potentials (below −70 mV). These channels remain inactive at resting membrane potentials (−60 to −65 mV) but deinactivate at hyperpolarized states, enabling calcium-driven LTS responses that generate $Na^+$-dependent action potentials. This results in an acceleration-deceleration bursting pattern, a hallmark of TRN neuron electrophysiology [18–20]. Abnormal rhythmic patterns involving T channels, such as spike-and-wave discharges, have been implicated in the generation and propagation of epileptic seizures, particularly absence seizures [21]. Moreover, TRN burst activity is thought to contribute to spontaneous oscillations within the nucleus and to function as a pacemaker for TC oscillations during sleep spindles and slow-wave sleep [14].

The expression of α7 nAChRs in TRN neurons has been demonstrated through mRNA detection and α-bungarotoxin ligand binding studies in both primates and rodents [14,22–27], underscoring their potential role in calcium-mediated cholinergic modulation of thalamic activity, as well as their involvement in epilepsy and sleep [28]. To investigate this, we developed a computational cell model that replicates the electrophysiological properties of TRN neurons, incorporating α7 nAChR-mediated calcium signaling [29,30]. Using this model, we examined how alterations in α7 nAChR expression affect TRN neuron firing. We extended our analysis to a simple TC network model capable of simulating wake- and sleep-like rhythms, in order to explorehow ectopic sleep-like rhythmic activity contributes to absence seizures.

## Materials and methods

### Single compartment TRN cell model

We developed a TRN cell model based on parameters established in published work [31–36]. Specifically, the basic firing kinetics of the model are based on the Hodgkin-Huxley formulation, which has been updated to include dynamic intracellular and extracellular ion concentrations [31–35]. To integrate α7 nAChR channel activity

into the TRN model, we used a modified model of the α7 nAChR including channel state transition as related to calcium conductance described in [36]. To model the distinctive electrophysiological properties of TRN neurons, we incorporated currents arising from T-type voltage-gated Ca²⁺ channels, Ca²⁺ dependent K⁺ channels, and Ca²⁺ dependent non-selective cation channels [37–39]. Parameters for α7 nAChR calcium coupling to the ER store through ryanodine (RyR) and inositol triphosphate receptors (IP₃R) are also described [36]. Deterministic formulations were used to classify the activity of RyR [40] and IP₃R [40–43] in a single compartment model where the uptake of Ca²⁺ through the SERCA pump in the ER is included [44]. Equations for ionic currents used in this model are provided in the **Supplementary** in File S1.

## Membrane dynamics of the TRN cell model

Differential equations that describe the single compartment cell model kinetics are listed below. Neuronal membrane potential ($V_m$) is modeled by the below equations with the unit's mV for voltage, µA/cm² for currents, mM for concentration, ms for time, and cm² for surface area. Hodgkin-Huxley equations are used to model the neuron's membrane potential, $V_m$:

$$\frac{dV_m}{dt} = -\frac{1}{C_M}\left(I_{Na} + I_K + I_{Cl} + I_T + C_M(I_{NCX}) + I_{KCa} + I_{\propto7} + I_{CAN} - I_{app}\right)$$

$$I_{Na} = G_{Na}\left(m_H^3\right)(h_H)(V_m - E_{Na}) + G_{NaL}(V_m - E_{Na})$$

$$I_K = G_K\left(j_H^4\right)(V_m - E_K) + G_{KL}(V_m - E_K)$$

$$I_{ClL} = G_{ClL}(V_m - E_{Cl})$$

$$I_T = G_{CaT}\left(m_{CaT}^2\right)(h_{CaT})(V_m - E_{Ca}) \tag{1}$$

$G_{Na}$, $G_K$, $G_{NaL}$, $G_{KL}$, $G_{ClL}$, $G_T$ represent Na⁺ voltage-gated conductance, K⁺ conductance voltage-gated conductance, Na⁺ leak conductance, K⁺ leak conductance, and Cl⁺ leak conductance, respectively. T-Type Ca²⁺ current is given by $I_T$ where $G_{CaT}$ is the maximum conductance of the Ca²⁺ current, $E_{Na}$, $E_K$, $E_{Cl}$, and $E_{Ca}$ are the reversal potentials, given by the Nerst Potential equation shown below. $I_{app}$ represents an externally applied current stimulus used to initiate network activity or simulate synaptic input. It is varied in time and amplitude depending on the experimental condition being modeled. The modified gating variable for T-type Ca²⁺ current, i.e., $m_{CaT}$ and $h_{CaT}$ are similar to those used for Na⁺ channel in the Hodgkin-Huxley model. The fraction of selective ions channel in the closed and open state is represented by the gating variables m, h, and n, which differ between 0 and 1. The opening and closing rates of the channel are also determined by $a_m$, $b_m$, $a_h$, $b_h$, $a_n$, $b_n$ as based on equations for sodium, potassium, and chloride leak current are obtained from [33]. We simulated the Na⁺-Ca2⁺ exchanger, $I_{NCX}$ using the formulation reported by [45]:

$$I_{NCX} = \left(\frac{I_{NCX_{max}}\left[\exp\left[\frac{(\gamma)VF}{RT}\right]\left[Na^+\right]_i^3\left[Ca^{2+}\right]_o - \exp\left[\frac{(\gamma-1)VF}{RT}\right]\left[Na^+\right]_o^3\left[Ca^{2+}\right]_i\right]}{(K_{m,Na}^3 + \left[Na^+\right]_o^3)(K_{m,Ca} + \left[Ca^{2+}\right]_o)(1 + k_{sat}\exp\left[\frac{(\gamma-1)VF}{RT}\right])}\right)$$

The Ca²⁺-dependent K⁺ current $I_{KCa}$ and Ca²⁺-dependent nonspecific cation current $I_{CAN}$ is based on a TRN cell model in [38]. Sodium and potassium currents $I_{Na}$ and $I_K$ including a leak current, are described in [33]. The α7 nAChR current is given by the equation below

$$I_{\propto7} = G_{\propto7}(V_m - E_{Ca})$$

where $G_{\propto7}$ is the maximal conductance across the α7 nAChR channel and $E_{Ca}$ is the reversal potential. Detail of the dynamic reversal potential for Na⁺ ($E_{NA}$), K⁺ ($E_K$), Cl⁻ ($E_{Cl}$), and Ca²⁺ ($E_{Ca}$), as well as the Ca²⁺ dyanamics equations, dynamic ion concentration equations, and parameter values, are discussed in the **Supplementary** in File S1. Differential equations were solved using ode15s in Matlab.

## Minimal network model

We incorpoted the single cell model into a simplified network where TC relay neurons receive GABAergic input from the TRN and display oscillatory bursting activity in response to TRN bursting [21]. TC neurons in turn excite the cortex and TRN through glutamatergic projections. TRN additionally receives excitatory feedback from corticothalamic projections [28]. GABAergic input onto the thalamic network from TRN cells results in a stereotyped excitatory feedback known as inhibitory rebound excitation that can be modeled as a time-delayed depolarizing current. A network model connecting the TRN cell to an excitatory feedback loop representing inhibitory rebound excitation in the thalamus was paired to inhibitory rebound excitation modeled as a depolarizing current. The feedback dynamics are controlled by parameters for the amplitude and duration, as well as a third parameter related to the delay time between TRN cessation and thalamic excitation. This delay time is related to the duration of the inhibition and is set by a recovery variable, y, obeying first order kinetics:

$$\frac{dy}{dt} = -\alpha y + \beta(1-y)$$

$\alpha$ is zero unless the membrane potential (Vm > zero), in which case it is a sufficiently large value to drive y close to zero during a single action potential. The constant $\beta$ determines the recovery time of y and is intended to model the duration of the GABA response [46]. The initiation of the thalamic rebound burst is triggered when the recovery variable reaches a value below 0.9. Finally, we incorporated a parameter controlling the strength of the potassium leak conductance, $G_{KL}$. The conductance is set between 0.07 and 0.21 uS/ms, resulting in resting membrane potentials between −64 and −80 mV. The effective change of the potassium leak conductance is intended to model adenosine activated channels as a means to control the wakefulness of the network. As this parameter affects the resting potential, it plays a key role in toggling between T-channel mediated tonic and burst firing.

Network reverberations were initiated in a TRN cell model using a 100 ms current injection, and interspike rates were recorded for 20 seconds post-stimulation. These rates were used to generate histograms, which were organized as heat maps to compare the frequency composition resulting from different model parameters. Parameters were looped in a nested structure based on their impact on network dynamics, transitioning from wake-like to sleep-like states, and were ordered along a "sleep-axis." Cross-correlation analyses quantified similarities between spike-rate histograms using Pearson correlation. A detailed description of the computational model and experimental setup is provided in **Supplementary** in File S1.

## Results

### Developing a TRN neuron model

The TRN is a shell of GABAergic neurons that surrounds the dorsal thalamus and sends projections to TC cells to form a reciprocal closed-loop circuit [47]. The TRN participates in regulating the flow of information through the thalamus and can influence processes related to epilepsy and sleep. The CHRNA7 gene, which encodes the α7 nAChR is expressed in thalamus. This gene is located on human chromosome15 within a site for 15q13mds neurodevelopmental disorders (**Fig 1A**). Combining published parameters for models of TRN neurons [38,39,48], as well as neuronal models featuring dynamic cellular ion concentrations [31,33,34], we developed a single compartment TRN neuron model that incorporates α7 nAChR channel calcium dynamics (**Fig 1B**). Primary currents within the cell model include voltage-gated fast $Na^+$, voltage-gated fast $K^+$, $Na^+$ leak, $K^+$ leak, $Cl^-$ leak, T-type voltage-gated $Ca^{2+}$, $Na^+$/$Ca^{2+}$ exchanger (NCX), $Ca^{2+}$-dependent $K^+$ $I_{KCa}$, and $Ca^{2+}$ -dependent nonspecific cation current $I_{CAN}$. Neuronal $K^+$ dynamics including uptake through glia and diffusion between the extracellular space and surrounding vasculature is also included as described in the dynamic equations (**Supplementary** in File S1). Neuronal pumps and cotransporters were also incorporated into the computational model, and this includes the neuronal $Na^+$/$K^+$-ATPase, glial $Na^+$/$K^+$-ATPase, and $K^+$/$Cl^-$ cotransporter (KCC2).

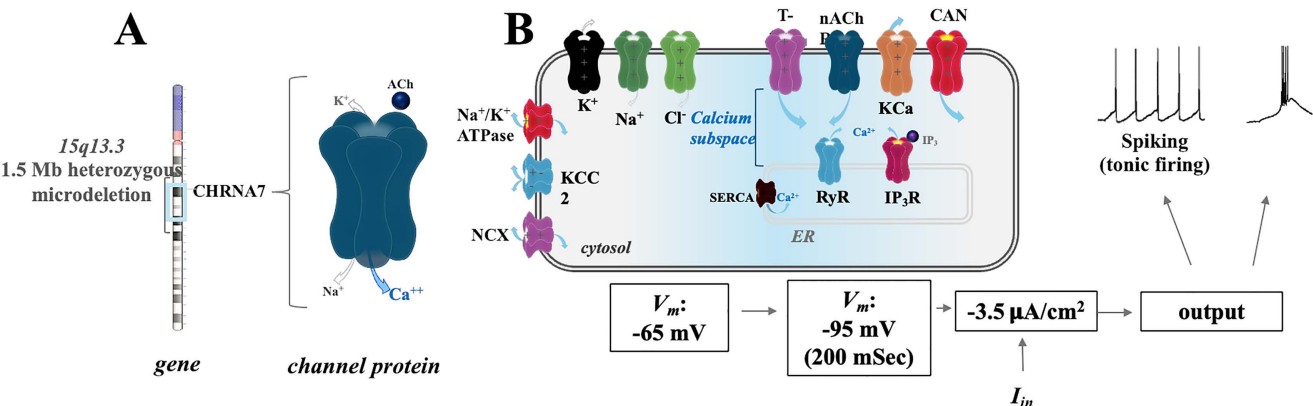

**Fig 1. Overview of the model.** (A) Schematic of human chromosome 15 and the location of CHRNA7. (B). A summary of the components of the TRN cell model. Abbreviations: Voltage-gated T-type calcium channel (T-type), α7 nicotinic acetylcholine receptor (nAChR), Calcium-dependent potassium current source (KCa), Calcium-activated nonspecific cation current source (CAN), Sodium-potassium pump (Na⁺/K⁺ ATPase), Potassium-chloride transporter member 5 (KCC2), Sodium-calcium exchanger (NCX), Sarcoendoplasmic reticulum calcium transport ATPase (SERCA), Ryanodine receptor (RyR), and Inositol 1,4,5-trisphosphate receptor (IP$_3$R).

The activity of neurons in the TRN depends on the voltage sensitivity of T-type channels [49] inactive at resting membrane potentials (−60 to −65 mV) and open at hyperpolarized potentials (−70 mV or lower) [20] resulting in a shift from neuronal tonic single-spike firing to low-threshold burst firing, respectively (**Fig 1**). T-type channels can also avoid de-inactivation in TRN neurons that are relatively hyperpolarized at resting membrane potential during sleep states, for example [50]. Our model takes into account ER calcium dynamics, where an intracellular Ca²⁺ triggers the activation of RyR channels located in the ER membrane (**Fig 1**).

Since T-type channels are typically inactive at resting membrane potentials (−60 to −65 mV) and open through de-inactivation at hyperpolarized potentials, we maintained the membrane resting potential at −69 mV. At 100 ms, we injected a hyperpolarized current of −3.5uA/cm² for 100 ms. This was found to trigger a burst response followed by spiking (**Fig 2**). Specifically, stimulation resulted in two distinct bursts with varying frequency and duration, and that was followed by an action potential train averaging 14 spikes (**Fig 2A**). This transition from bursting to spiking is illustrated in the magnified portion of the action potential trace (**Fig 2B**), showing a pattern consistent with the accelerando–decelerando bursting activity, as previously reported by Fuentealba et al. [51]. Neuronal responses were found to be temporally coupled to a de-inactivation of T-type channels. Here, burst responses correlated with a strong but diminishing I$_T$ while the spike activity was associated with repeated low-level I$_T$ less then 50% of the original values (**Fig 2C**). Modifications to intracellular calcium (Ca²⁺$_i$) as well as localized calcium concentration (Ca²⁺$_{local}$) were assessed as described [36]. As shown in **Fig 2D and 2E**, [Ca²⁺]$_i$ and [Ca²⁺]$_{local}$ transients were observed in a manner consistent with the membrane potential firing response of the neuron as well as the de-inactivation of T-type channels in the model.

## A role for α7 nAChR calcium in TRN neuron responses

To test the role of α7 nAChR expression, we removed the receptor response from equation parameters within the cell model producing an "-α7" model. Simulations were performed comparing -α7 to the control baseline model using a −3.5uA/cm² hyperpolarizing current injection for 100 ms. As shown in **Fig 3**, a loss in α7 nAChR was associated with the absence of TRN neuron tonic firing (spiking) and a modification in the bursting response. Specifically, the model reveals an effect of α7 nAChR on the onset and the duration of the second burst with -α7 simulation exhibiting a 17 ms delay and a 56 ms reduction in the burst duration compared to the control, which was followed by tonic firing. This effect appears

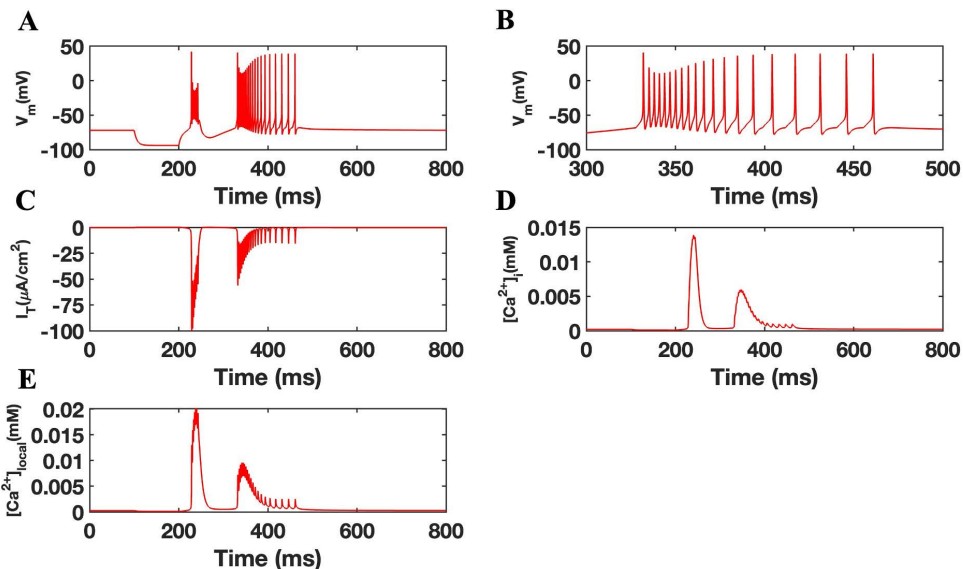

**Fig 2. Activation and firing properties of the TRN cell model.** (A) Bursting behavior followed by tonic firing in control simulations of a TRN cell model after the injection of a stimulating current. (B) Zoomed-in view of the transition from bursting to tonic spiking, highlighting the accelerando–decelerando structure of the burst followed by sustained spiking. (C) Membrane T-type current. (D) Intracellular calcium concentration. (E) Calcium concentration in the local subspace.

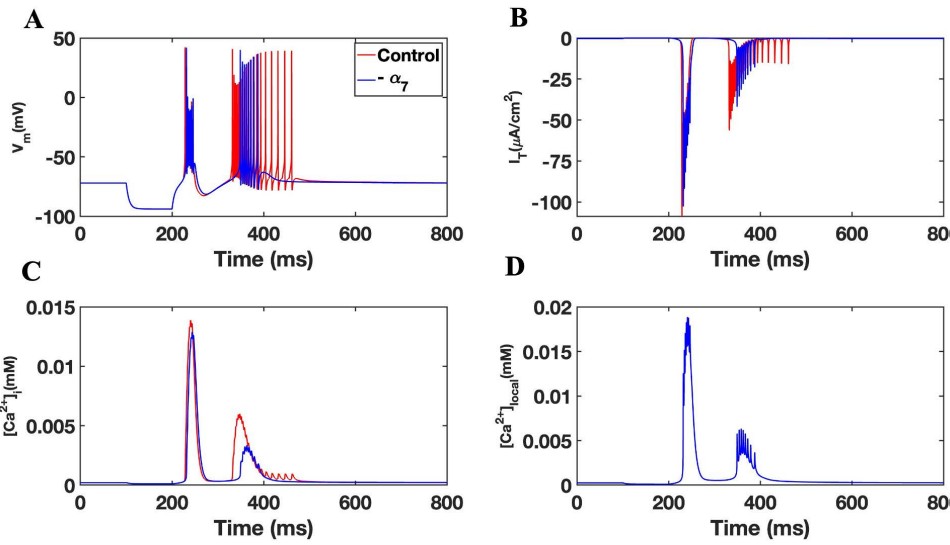

**Fig 3. Involvement of the nAChR in TRN cell firing.** (A) Bursting accompanied by tonic firing in TRN simulations in the presence (red) and absence (blue) of the α7 nAChR. (B) Membrane T-type current. (C) Intracellular calcium concentration. (D) The concentration of calcium in the local subspace.

coupled to a loss in tonic firing within -α7 (**Fig 3A**). These findings indicate that α7 nAChR activity contributes to the temporal structure of burst firing and the transition to tonic spiking in TRN neurons.

Neuronal responses in the -α7 experiment were also temporally coupled to a de-inactivation of T-type channel responses (**Fig 3B**). Here however a loss in α7 nAChR was associated with a slight delay in the $I_T$ that is matched to

the second burst and shows that modified bursting activity coincides with a diminished $I_T$ in the cell model. This result is consistent with alteration in $[Ca^{2+}]_i$ as well as membrane $[Ca^{2+}]_{local}$ within -α7. As shown in **Fig 3C**, $[Ca^{2+}]_{local}$ transients were attenuated in - α7 compared to the control with a loss in the duration of the calcium transient response at points that correlate with the neuron spiking activity and $I_T$ response. A loss in α7 nAChR appears to modify the impact of the hyperpolarizing stimulus input on both the firing and calcium signaling within the TRN neuron.

The activation of the α7 nAChR is associated with ER store calcium release through calciun induced calcium release (CICR) [36]. We tested the involvement of RyR and IP$_3$R in our TRN neuron model. A hyperpolarized current of −3.5uA/cm² for 100 ms was used to stimulate TRN neuron responses as shown earlier. An analysis relating state transitions of RyR and IP$_3$Rs from closed to open state was conducted as shown previously [36]. As shown in **Fig 4**, TRN neuron bursting and tonic firing was correlated with the activation of RyR and a slower activation of IP$_3$Rs. Specifically, **Fig 4C** shows no flux from IP$_3$R during the first burst depicted in **Fig 4A** while IP$_3$R activity is observed during the second and third bursts. These results are consistent with expected the sequential binding of IP$_3$ and Ca²⁺ ligands to IP$_3$R and RyR, respectively [41,44]. In this experiment, blocking of RyRs and IP$_3$Rs (by keeping them in the closed state or multiplying their fluxes by zero) did not change the bursting behavior of the second spike nor play a role in the eradication of the tonic tail single spikes (**Fig 5**).

## Neural network dynamics and the sleep-axis

The TC network is implicated in the pathophysiology of epilepsy as well as sleep and directly modulated by ACh input that originates from basal forebrain and brain stem (**Fig 6**). We examined the hypothesis that absence-type seizures are a form of ectopic sleep-like rhythm, through an analysis of wake- and sleep-like rhythms generated by our network model under different levels of α7 nAChR expression within TRN. First, we investigated the interspike rate histograms (**Fig 7**) for network responses to the same stimuli with normal α7 nAChR expression, but with different adenosine, and feedback parameters. The results are ordered alonga "sleep-axis" where the parameters for adenosine conductance,rebound delay,

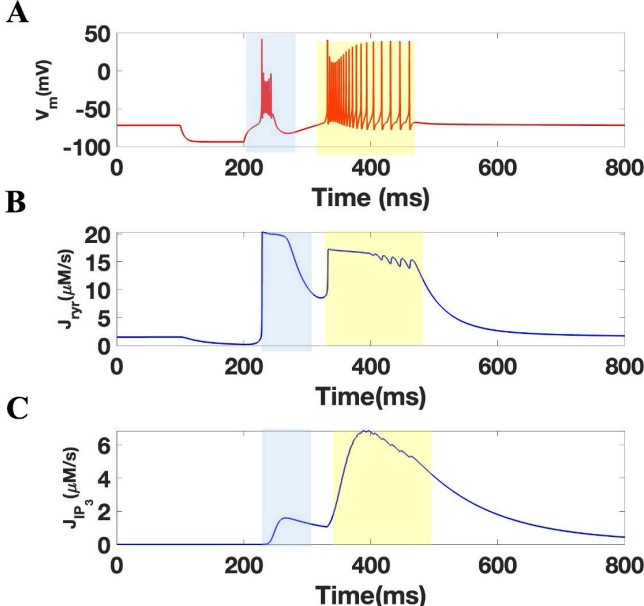

**Fig 4. ER channel calcium flux during TRN activity.** (A) The membrane potential of TRN cell during calcium flux through RyR (B) and IP$_3$R (C). The blue and yellow highlight indicates regions of bursting and tonic firing, respectively.

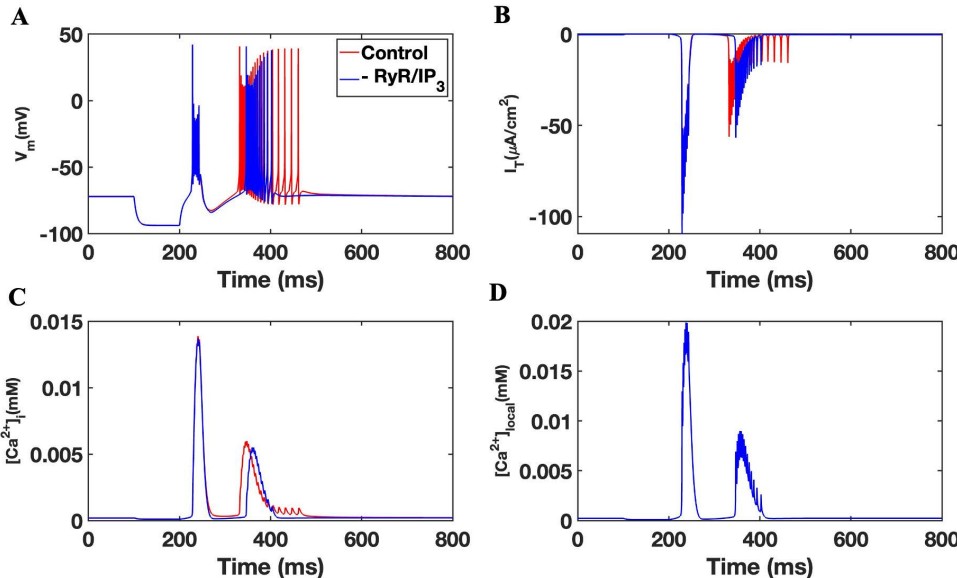

**Fig 5. TRN cell firing is dependent on calcium store activity.** (A) Membrane Potential (B) Membrane T-type current, (C) Intracellular calcium concentrations and nearby subspace (D). Simulations with (blue) and without (red) block of RyR and IP$_3$R.

and strength and duration respectively, are cycled based on their relative impact on the network. As shown, the inner most loop shows the smallest change, namely the rebound duration, and the outer most loop consists of the parameter with the most influence showing adenosine level.

In **Fig 7** we display interpike rate histograms over frequency ranges of 0.01 to 380 Hz (top row) and 0.01 to 45 Hz (bottom row) for all panels. The frequency content for baseline (100%) α7 nAChR expression is presented in **Fig 7B**. Adenosine levels are fixed at normal, intermediate and high levels along the first, second and last third of the sleep-axis respectively. The dynamics within the initial third of the sleep-axis are all characterized by a consistent and restricted island of interspike rates between 15 and 30 Hz with no low frequency components.

In comparison, the states in the middle third are described by slow interburst frequencies between 1 and 17 Hz (black circles) in the lower panel, and faster intraburst rates ranging from 40 to 200 Hz (black ovals) in the upper panel (**Fig 7B**). The last third of the parameters produce low 1 and 17 Hz interburst type frequencies, like the middle third, but with fewer high frequency components. The dynamics within this intial region, low-adenosine, are highly distinguishable from the intermediate and high adenosine levels. They also posses a stereotypical response of intermediate frequency components with no low (<10 Hz) components and suggestive of wake states. The remaining regions contain significant low frequency components with the highest adenosine level containing a number of transient responses. Examples of the time traces and individual interspike rate histograms for these different wake-sleep states are displayed in **Fig 7D**. Wake-like states are shown in the top two panels, and bursting and transient sleep-like states shown in the bottom four.

The two panels of **Fig 7A** show the results of a 50% reduced α7 nAChR expression. As shown the middle and right third, i.e., sleep-like states, are similar to the sleep-like states in the 100% α7 expression case (**Fig 7B**), especially in burst frequency. However, with reduced α7 nAChR theintraburst spiking appears less well defined and is distributed over a wider range of frequencies. The biggest difference between normal and reduced expression, is found in the change to the wake-like states existing in the first third of the sleep-axis. At 100% expression, there are no burst-like dynamics and a a well-defined island of activity from 15–30 Hz. In contrast, at 50% expression, the island is nearly absent and there now appears burst-like states that are akin to the sleep-like states seen for both levels of expression.

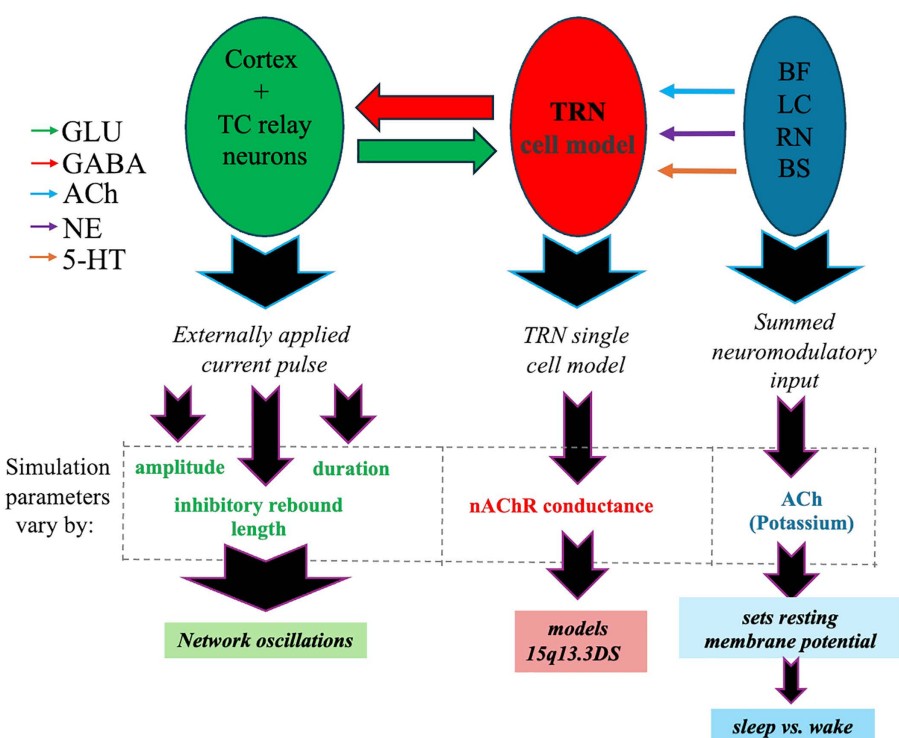

**Fig 6. Schematic overview of the TC neural network model used to simulate oscillatory dynamics.** The model includes TC relay neurons that receive inhibitory GABAergic input from the TRN cell model and send excitatory glutamatergic output to the cortex. Neuromodulatory input to the TRN originates from: basal forebrain (BF; cholinergic input), locus coeruleus (LC; noradrenergic input), raphe nuclei (RN; serotonergic input), and brainstem (BS; mixed modulatory input including cholinergic and monoaminergic signals). These inputs modulate TRN excitability and set the resting membrane potential. Simulation parameters: input amplitude, duration, inhibitory rebound length, and α7 nAChR conductance, which are varied to model sleep–wake transitions and 15q13.3mds. The timing of feedback between TRN cessation and thalamic excitation regulates network oscillation frequency, distinguishing between physiological sleep spindles and absence seizure-like activity..

To further elucidate the contribution of α7 nAChR into sleep-like transition, we ran the model with α7 expression set at multiple expression levels: 0, 75, 90, 110, and 125%. These results are displayed in **Fig 7C** over the same frequency ranges in the top and bottom panels of **Fig 7A** **and** **7B**. The most remarkable effect, seen in the lower panels of **Fig 7C**, is the shifting of wake-like island to slower rhythms as indicated by the black arrow as the α7 expression is decreased. As α7 nAChR expression falls below 90%, sleep-like burst frequencies start to emerge in the previously wake-like region, coexisting with slower, but tonic wake-like states. As α7 nAChR expression is reduced to 50%, the island has dissipated and the dynamics are nearly all sleep-like.

Correlation coefficients between the interspike rate histograms were used to identify sleep- and wake-like states for different model parameters (e.g., α7 level, inhibition feedback, adenosine level). These results display a grid of the correlations across all parameters with the heat map indicating the degree of correlation (**Fig 8A**). Fig 8C is an expanded view of the green box for 100% α7 nAChR expression within panel A, and shows a clear distinction between low adenosine (light gray box) and either intermediate (dark gray) or high adenosine states (black box). For the intermediate and high adenosine, dark gray and black boxes respectively, the correlations are weaker and less uniform than under wake-like conditions. In addition, significant overlap between the two higher levels of adenosine are indicated by the bright regions below and to the right of the central gray box and suggest similar dynamics between them.

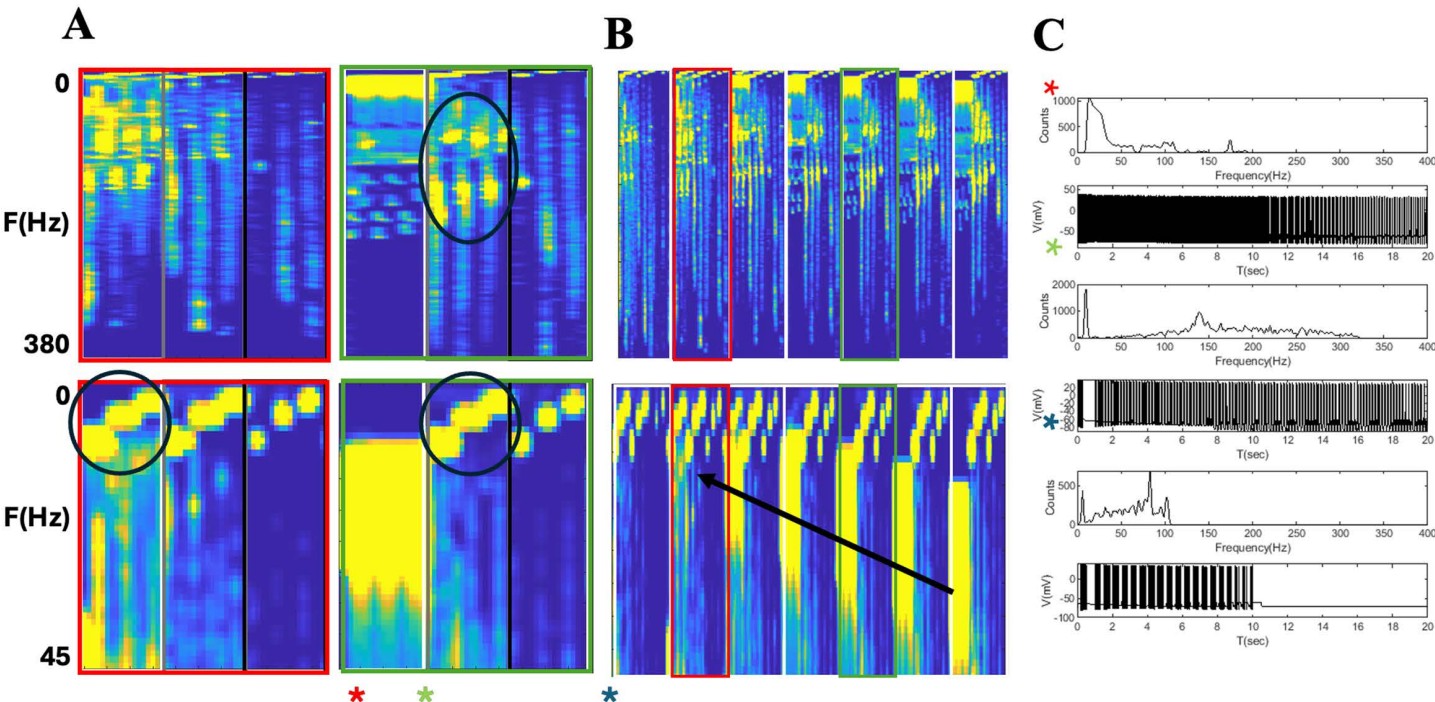

**Fig 7. Frequency composition of TC network reverberations across model parameters.** (A–B) A display of the parameter spaces constructed from the interspike rate histograms. The two left figures (A) are the parameters spaces for the 50% expression case and the two on the right are for normal 100% expression. The top figures range from 0.01 to 380 Hz, and the bottom figures go to 45 Hz. (B) The space for the power spectra are displayed for a larger range:0, 50,75, 90, 100, 110, 125%. (C) Time traces and individual interspike rate histograms for three distinct dynamics: tonic, fast bursting, and transient slow bursting. The colored asterisks indicate their location in parameter space.

**Fig 8B** provides an expanded view of the 50% expression region denoted by the red box in panel A of the same figure. Here the region of low adenosine (light gray box) displays far less correlations than seen in Fig 8C. Furthermore correlations between low and intermediate adenosine concentrations are nearly as high as the low-low correlations and the low to high state even share some regions of strong correlation. Along with the results from **Fig 7** that demonstrate the emergence of bursting states in the normally tonic wake state, the results here show that sleep-like states exist for all adenosine levels under partial α7 nAChR expression. White horizontal rectangles intersect the low adenosine regions of the normal, bottom, and 50% expression, top, in **Fig 8A**. For the normal case, in the lower rectangle, the wake-like states for 90, 100, 110 and 125% nAChR expression all share high correlations, while the 75, 50, and 0% expression show diminishing correlations with the normal state. Finally, the upper white rectangle shows that the correlations between expected wake states for the 50% expression case demonstrate high correlaions not only with their own sleep states, but the sleep states of the normal and high α7 nAChR expression, demonstrating, again, the need for α7 nAChR expression for wake-sleep discrimination in our model.

## Discussion

Individuals with 15q13.3mds exhibit a strong risk for recurrent seizures with approximately one third of the impacted population exhibiting childhood onset absence seizures [10]. Earlier studies support the role of nAChR signaling in the development of seizure activity and the etiology of epilepsy. In addition to CHRNA7, several nAChR genes have been implicated in seizure disorders including CHRNA2, CHRNA4, CHRNB2 [52]. CHRNA7 however is directly impacted in

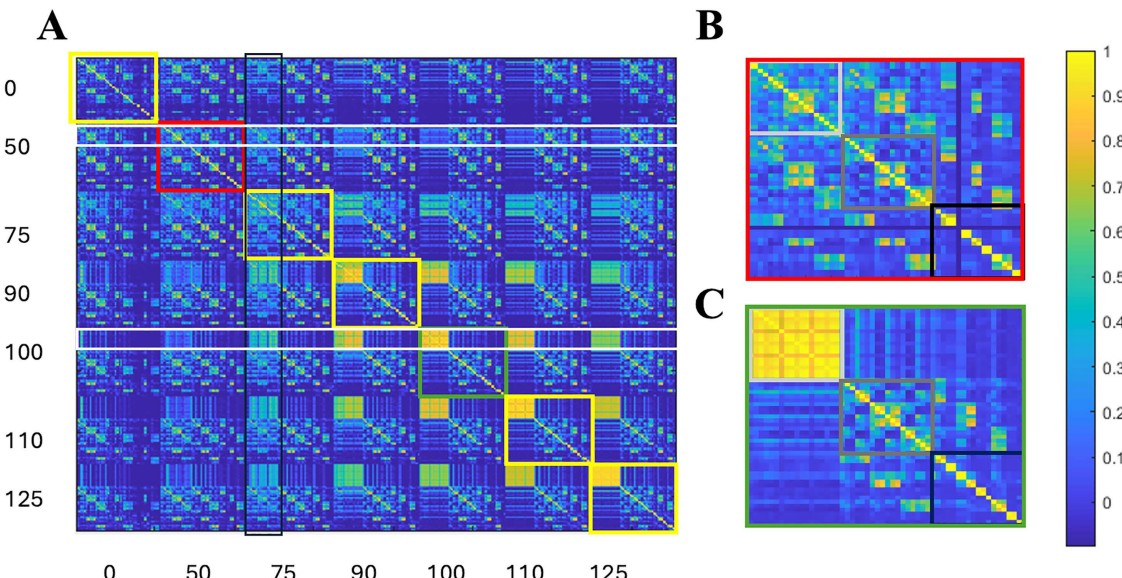

**Fig 8. Identifying sleep and wake-like states through correlation analysis of network dynamics across model parameters.** (A) The cross correlation coefficients for inter-spike histograms for α7 nAChR conductance at 0, 50, 75, 90, 100, 110, and 125%. The 50% (red box) and 100% (green box) insets are expanded in (B) and (C).

15q13mds with a loss of one copy of the gene shown to reduce α7 nAChR tissue expression likely leading to cholingeric system related cognitive impairment, seizure occurance, and neuropsychiatric symptoms. Because α7 nAChRs are highly permeable to $Ca^{2+}$, changes to the expression of this receptor are expected to impact CICR as well as IICR through PLC coupled G protein signaling [36] during TRN neuronal firing [53]. For example, the administration nicotine contributes to the induction of tonic-clonic seizures in mice through α7 nAChR activity [54,55]. A mutation in the CHRNA7 gene appears to promote susceptibility to nicotine-induced seizures in mice [56]. How changes in α7 nAChR expression or function within neurodisorders such as 15q13.3mds may drive the development of seizure activity is not clear.

Changes in $Ca^{2+}$ signaling within TRN neurons are known to contribute to absence seizure generation through the regulation of T-type channels in various models including rodent genetic absence epilepsy rat from strasbourg (GAERS) [57]. Cholinergic modulation of T-type channel function within TRN can also impact TC activity, leading to epileptic seizures, while also contributing to the generation and maintenance of different sleep stages. Our computational experiments support this notion and highlight the role of α7 nAChR calcium signaling in TRN neuronal firing, and in feedback of regulatory circuits involved in TC oscillations, based on intracellular ER and T-type channel modutional. Interactions between T-type channels and nAChRs have been shown in other cells including neuroendocrine cells contributing to catecholamine release [58]. Mechanistically, our single compartment model highlights the role of coupling between intracellular calcium stores and T-type channel activity as previously shown in rat brain slices [59]. In addition, we show an impact of the nAChR on other membrane currents including SK2 channel driven $Ca^{2+}$ dependent $K^+$ and $Na^+/Ca^{2+}$ exchanger (NCX) resulting in changes to burst length of TRN neurons and a loss of tonic firing within cells that lack α7, respectively.

Neuronal activity bursts are a reliable way to send a high-efficacy signal to a postsynaptic target and are a suitable mode for signal detection when compared to spikes in tonic mode [60]. We sought to replicate the *accelerando-decelerando* bursting activity seen experimentally based on our adaptation of a single compartment TRN model [38]. This model displays two main types of bursting activity: repeated bursts when the conductance for the $Ca^{2+}$-dependent nonspecific cation current ($I_{CAN}$) is set to 0, and an *accelerando-decelerando* pattern of bursting followed by a tonic tail of individual spikes

when $I_{CAN}$ is included. $I_{CAN}$ involvement in a burst sequence leads to an afterdepolarization (ADP) effect, which over time slows the firing rate within individual bursts, resulting in an inactivation of $I_T$ and a tonic tail of single spikes similar to what is seen *in vivo* [37]. Our model replicates the accelerando–decelerando burst activity that is observed experimentally, with a burst response triggered by hyperpolarizing current injection exhibiting a progressive increase in inter-spike frequency (accelerando phase) followed by a gradual slowing down (decelerando phase). This temporal structure closely mirrors the intrinsic firing behavior of TRN neurons [51] and demonstrates that our model reproduces key activity dynamics of TRN neurons.

## Cholineric modulation of network activity, sleep, and epilepsy

One theory regarding the mechanism of absence seizures involves the concept of an ectopic sleep-like rhythm proposing that the TC circuits responsible for generating sleep rhythms become activated inappropriately during wakefulness, leading to the characteristic features of absence seizures. Ascending cholinergic projections (from basal forebrain) exert control of forebrain excitability, and wide evidence implicates mAChR and nAChR in epileptiform activity [61]. Cholinergic nuclei are mostly active during wakefulness and rapid-eye-movement (REM) sleep, but almost silent during non-REM sleep [62]. Cholinergic modulation constitutes an important regulator of the cortical tone contributing to states of arousal, attention, and cognitive performance during wakefulness. Tonic-clonic seizures are triggered by administration of high doses of nicotinic agonists, whereas non-convulsive doses have kindling effects in mice [63].

Our results indicate a role for α7 nAChRs in the switch between sleep and wake-like dynamics within the TRN thalamic circuit. Our network simulations show that without the α7 nAChR conductance, or even in its reduced state, the delimination between wake and sleep-like states is greatly degraded and the existence of what are expected to be normal wake-like states are no longer available to the network.

Feedback loops involving the TRNand corticothalamic neurons generate TC oscillations. Our model indicates that bursting activity of TRN neurons, including the duration and frequency of each burst, is crucial to determining the frequency and synchrony of TC oscillations, which are key factors in distinguishing between sleep spindles and pathological absence seizure activity. Specifically, TC oscillations giving rise to absence seizures, that are defined by frequency and level of synchrony with absence seizures in humans, are characterized by 3 Hz oscillations (spike and wave discharges) contrast with 11–16 Hz oscillations that define sleep spindles, a marker of non-REM sleep states [18]. Our minimalistic network model successfully produces bursts at both of these frequencies for sleep-like states, with normal nAChR expression, and also under wake-like conditions for the reduced nAChR expression cases. In this model we observe that the slower bursting frequency is determined by the duration of the inhibitory feedback delay. Bursting in TRN neurons helps define these oscillatory states, compared to tonic firing present in TRN neurons during the awake state. This switch between bursting and firing modes is mediated by the presence of T-type voltage gated calcium channels in the TRN [19] and by an adenosine dependent leak channel. This framework could be further tested in future experiments using other mechanisms, such as mAChR and norepinephrine targeting that are known to be important for sleep transition [16].

In conclusion, our computational model of reduced CHRNA7 gene expression provides insight into the role of α7 nAChRs in thalamic seizure activity, particularly through their impact on $Ca^{2+}$ signaling during TRN neuronal firing. Our findings suggest that altered α7 nAChR activity can disrupt TRN firing, leading to a loss in the distinction between wake and sleep-like states, and may contribute to the generation of absence seizures in human disorders like 15q13.3mds. Cellular interaction between α7 nAChRs, T-type channels, and CICR appears to play a role in TRN bursting and contribute to TC oscillations. By modeling these interactions, we propose a mechanism by which cholinergic modulation can both promote and disrupt normal sleep rhythms, with potential implications for understanding the pathophysiology of epilepsy and sleep-related disorders. Our results underscore the importance of the human α7 nAChR in maintaining the balance of network activity that maybe of value for therapeutic strategies for the treatment of epilepsy and sleep disorders.

## Supporting information

**File S1. Supplementary Methods: Computational Model Experiment.** Table 1 – Parameters. Table 2 – Initial Values. (DOCX)

## Author contributions

**Conceptualization:** John R. Cressman, Nadine Kabbani.

**Data curation:** Aman Ullah, Karen Therrien, John R. Cressman, Nadine Kabbani.

**Formal analysis:** Aman Ullah, Karen Therrien.

**Investigation:** Aman Ullah.

**Methodology:** Aman Ullah, Nadine Kabbani.

**Software:** Aman Ullah, Karen Therrien, John R. Cressman.

**Supervision:** Aman Ullah, John R. Cressman, Nadine Kabbani.

**Visualization:** Aman Ullah, John R. Cressman.

**Writing – original draft:** Karen Therrien.

**Writing – review & editing:** Aman Ullah, John R. Cressman, Nadine Kabbani.

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
