## [Decision Letter · Decision Letter 0]

22 Apr 2025

PONE-D-25-14714Computational model of alpha 7 nicotinic acetylcholine receptor in thalamic reticular nucleus neurons and their involvement in network statesPLOS ONE

Dear Dr. Ullah,

Thank you for submitting your manuscript to PLOS ONE. After careful consideration, we feel that it has merit but does not fully meet PLOS ONE’s publication criteria as it currently stands. Therefore, we invite you to submit a revised version of the manuscript that addresses the points raised during the review process.

We look forward to receiving your revised manuscript.

Kind regards,

Giuseppe Biagini, MD

Academic Editor

PLOS ONE

Reviewers' comments:

Reviewer's Responses to Questions

**Comments to the Author**

1. Is the manuscript technically sound, and do the data support the conclusions?

Reviewer #1: Partly

2. Has the statistical analysis been performed appropriately and rigorously? 

Reviewer #1: N/A

3. Have the authors made all data underlying the findings in their manuscript fully available?

Reviewer #1: Yes

4. Is the manuscript presented in an intelligible fashion and written in standard English?

Reviewer #1: Yes

5. Review Comments to the Author

Reviewer #1: The manuscript by Ullah et al. presents a single-compartment Hodgkin-Huxley (HH) model of a TRN neuron incorporated into a simplified thalamic network model, aiming to demonstrate that alpha7 nAChR activity modulates TC oscillations. Overall the scientific questions are clearly defined and the methodological approach is correct. However, there are several major issues that need to be addressed before the manuscript can be accepted for publication.

Here are my comments:

MAJOR CONCERNS

1) The Materials and Methods section lacks sufficient detail, and the overall presentation of the results is difficult to follow. For example, the current input (Iapp) in the first set of equations is not explained, and the inclusion of the Nernst equation seems redundant and can be omitted to include other fundamental elements (see below). The authors should provide a more thorough and clear presentation of the materials and methods.

2) The authors claim to have constructed a neural network consisting of a single TRN neuron and an unspecified number of TC relay neurons. Can the dynamic description of an inhibitory rebound excitation be strictly considered a neural network model? Please comment more about your assumptions.

3) In the "Neural Network Model" section, the authors reports: "........Finally, we incorporated a parameter controlling the strength of a potassium leak current modeling adenosine activated channels as a means to control the wakefulness of the network. This final parameter affects the resting potential, thus playing a key role in toggling between T-channel mediated tonic and burst firing.........."

However, the name of this parameter and its modulation range is neither mentioned in the main text nor in the supplemental material. Please provide a complete description of the model, including this missing information. Since the work is entirely based on modeling, it is essential that all model parameters are fully described.

4) In the Results section, under "Neural Network Dynamics and the Sleep Axis," a description of the network is provided. This network description should be included earlier in the Methods section, which currently lacks essential information necessary to understand how the network is conceptualized and implemented. Most importantly the explanation of the sleep axis isn't clear. A dedicated section should be included in the Methods, outlining this aspect along with an explanation of the criteria used to assess the impact of parameters on the network's behavior.

5) The explanation of the Results and the related figures are not sufficiently clear. Figure 7 and 8 are poorly commented.

Fig.7: How is the spectral analysis performed? What does the count plot represent? The colorbar of the heatmap is missing together with an adeguate description of plot axes and parameter changes. What do the black circles represent? Indicating the decreasing of the alpha7 with a black arrow is not straightforward.

The authors write "White horizontal rectangles intersect the low adenosine regions of the normal, bottom, and 50% expression, top, in Fig. 7A", where are the white horizontal rectangles in the figure?

Fig 8: How is the cross-correlation performed?

6) "Here, the dynamics between wake, the first horizontal third, differs from the sleep states, middle and right thirds, in several remarkable ways." This sentence is impossible for me to decipher, and the significant differences are not explicitly discussed.

7) Does the model effectively replicate the accelerando-decelerando burst activity observed experimentally? This evidence should be thoroughly highlighted and discussed in the Results section, rather than being confined to the Discussion.

MINOR

1) There are several minor typos that need to be corrected. For example:

"nonselective"

"INCXusing"

"activitybursts"

A careful review of the English is highly recommended.

2) The legend of Fig.6 depicts the neural network but the legend doesn't explain all the elements of the figure (what is BF, LC, RN, BS?). Once that the figure layout has been improved please check carefully the legend description that must match the figure content.

6. PLOS authors have the option to publish the peer review history of their article (what does this mean? ). If published, this will include your full peer review and any attached files.

**Do you want your identity to be public for this peer review?** For information about this choice, including consent withdrawal, please see our Privacy Policy .

Reviewer #1: No

---

## [Author Response · Author response to Decision Letter 1]

15 Jul 2025

We thank the Editor and Reviewers for their careful reading of our manuscript and for the thoughtful comments and suggestions. We have addressed each point in a reviewer response file.

---

## [Decision Letter · Decision Letter 1]

5 Aug 2025

Computational model of alpha 7 nicotinic acetylcholine receptor in thalamic reticular nucleus neurons and their involvement in network states

PONE-D-25-14714R1

Dear Dr. Ullah,

We’re pleased to inform you that your manuscript has been judged scientifically suitable for publication and will be formally accepted for publication once it meets all outstanding technical requirements.

Kind regards,

Giuseppe Biagini, MD

Academic Editor

PLOS ONE

Additional Editor Comments (optional):

Reviewers' comments:

Reviewer's Responses to Questions

**Comments to the Author**

1. If the authors have adequately addressed your comments raised in a previous round of review and you feel that this manuscript is now acceptable for publication, you may indicate that here to bypass the “Comments to the Author” section, enter your conflict of interest statement in the “Confidential to Editor” section, and submit your "Accept" recommendation.

Reviewer #1: All comments have been addressed

2. Is the manuscript technically sound, and do the data support the conclusions?

Reviewer #1: Yes

3. Has the statistical analysis been performed appropriately and rigorously? 

Reviewer #1: N/A

4. Have the authors made all data underlying the findings in their manuscript fully available?

Reviewer #1: Yes

5. Is the manuscript presented in an intelligible fashion and written in standard English?

Reviewer #1: Yes

6. Review Comments to the Author

Reviewer #1: The authors have satisfactorily addressed my comments, and I recommend the article for publication. I suggest doing a final check for typos

7. PLOS authors have the option to publish the peer review history of their article (what does this mean? ). If published, this will include your full peer review and any attached files.

**Do you want your identity to be public for this peer review?** For information about this choice, including consent withdrawal, please see our Privacy Policy .

Reviewer #1: No

---

## [Editor Report · Acceptance letter]

PONE-D-25-14714R1

PLOS ONE

Dear Dr. Ullah,

I'm pleased to inform you that your manuscript has been deemed suitable for publication in PLOS ONE. Congratulations! Your manuscript is now being handed over to our production team.

Kind regards,

on behalf of

Dr. Giuseppe Biagini

Academic Editor

PLOS ONE